# Investigation of the Relationship between Lean Muscle Mass and Erythropoietin Resistance in Maintenance Haemodialysis Patients: A Cross-Sectional Study

**DOI:** 10.3390/ijerph19095704

**Published:** 2022-05-07

**Authors:** Wen-Fang Chiang, Po-Jen Hsiao, Kun-Lin Wu, Hung-Ming Chen, Chi-Ming Chu, Jenq-Shyong Chan

**Affiliations:** 1Division of Nephrology, Department of Medicine, Armed Forces Taoyuan General Hospital, Taoyuan 325, Taiwan; wfc96076@hotmail.com (W.-F.C.); ndmc6217316@yahoo.com.tw (K.-L.W.); 2Division of Nephrology, Department of Medicine, Tri-Service General Hospital, National Defense Medical Center, Taipei 114, Taiwan; 3School of Medicine, National Defense Medical Center, Taipei 114, Taiwan; 4Department of Life Sciences, National Central University, Taoyuan 320, Taiwan; 5Division of Nephrology, Department of Medicine, Fu-Jen Catholic University Hospital, School of Medicine, Fu-Jen Catholic University, New Taipei City 242, Taiwan; 6Division of Haematology and Oncology, Department of Medicine, Armed Forces Taoyuan General Hospital, Taoyuan 325, Taiwan; mptt@aftygh.gov.tw; 7Graduate Institute of Life Sciences, National Defense Medical Center, Taipei 114, Taiwan; chuchiming@web.de; 8School of Public Health, National Defense Medical Center, Taipei 114, Taiwan; 9Graduate Institute of Medical Sciences, National Defense Medical Center, Taipei 114, Taiwan; 10Department of Public Health, School of Public Health, China Medical University, Taichung 404, Taiwan; 11Department of Public Health, Kaohsiung Medical University, Kaohsiung 807, Taiwan; 12Big Data Research Center, Fu-Jen Catholic University, New Taipei City 242, Taiwan; 13Division of Biostatistics and Medical Informatics, Department of Epidemiology, School of Public Health, National Defense Medical Center, Taipei 114, Taiwan

**Keywords:** anaemia, bioimpedance spectroscopy, erythropoietin resistance, haemodialysis, haemoglobin, lean muscle mass, sarcopenia

## Abstract

Each patient undergoing maintenance haemodialysis (MHD) has a different response to erythropoiesis-stimulating agents (ESAs). Haemodilution due to fluid overload has been shown to contribute to anaemia. Body mass index (BMI) has been shown to influence ESA response in dialysis patients; however, BMI calculation does not distinguish between fat and lean tissue. The association between lean muscle mass and erythropoietin hyporesponsiveness is still not well-known among MHD patients. We designed a cross-sectional study and used bioimpedance spectroscopy (BIS) to analyse the relationship between body composition, haemoglobin level, and erythropoietin resistance index (ERI) in MHD patients. Seventy-seven patients were enrolled in the study group. Compared with patients with haemoglobin ≥ 10 g/dL, those with haemoglobin < 10 g/dL had higher serum ferritin levels, malnutrition–inflammation scores (MIS), relative overhydration, ESA doses, and ERIs. In multivariate logistic regression, higher ferritin levels and MIS were the only predictors of lower haemoglobin levels. The ERI was significantly positively correlated with age, Kt/V, ferritin levels, and MIS and negatively correlated with albumin levels, BMI, and lean tissue index (LTI). Multivariate linear regression analysis revealed that ferritin levels, BMI, and LTI were the most important predictors of ERI. In MHD patients, using BIS to measure body composition can facilitate the development of early interventions that aim to prevent sarcopenia, support ESA responsiveness, and, consequently, improve anaemia management.

## 1. Introduction

Anaemia is a common manifestation of chronic kidney disease (CKD) because of the reduced production of erythropoietin by the diseased kidneys. As kidney function deteriorates, haemoglobin levels progressively decrease, which is particularly evident occurrence in CKD patients undergoing dialysis. In addition to several adverse consequences, including exacerbations of angina, left ventricular hypertrophy, deranged haemostatic function, and impaired immune function, anaemia can contribute to an increased risk of morbidity and mortality in patients with end-stage renal disease (ESRD) [1].

Multiple underlying mechanisms of anaemia exist in patients undergoing maintenance haemodialysis (MHD). In brief, a low haemoglobin level may result from haemodilution caused by an increased extracellular volume (ECV) and from true anaemia caused by decreased red blood cell mass [2,3]. Therefore, the evaluation of anaemia in MHD patients must include an investigation of underlying causes that lead to decreased erythrocyte production, haemolysis, and blood loss, as well as a fluid overload status assessment. ECV overload is a common issue in MHD patients and is associated with mortality [4]. However, precise quantification of fluid status in dialysis patients remains a challenge in clinical practice. The gold standard method of measuring ECV is bromide dilution, but this method is expensive and time consuming [5]. Lung ultrasound and measurement of inferior vena cava diameter can be used to evaluate fluid overload; however, their use is limited by the availability of trained personnel and operator dependency. Right heart catheterisation provides measurements of haemodynamic data, including right atrium pressure, pulmonary arterial pressure, and pulmonary capillary wedge pressure, and thus can be an indicator of volume status. However, it is an expensive and invasive procedure requiring trained personnel and may be uncommonly complicated with carotid artery injury, arteriovenous fistula formation, and tricuspid valve injury [6]. Bioimpedance methods utilising either single-frequency bioimpedance analysis (mostly at a frequency of 50 kHz) or multifrequency bioimpedance spectroscopy (BIS, at frequencies ranging from 5 to 1000 kHz) are simple and noninvasive methods that have been validated as reliable tests for the assessment of body fluids in dialysis patients [7]. However, there have been few studies investigating fluid status in relation to anaemia in MHD patients.

Although hypoxia-inducible factor stabilisers have demonstrated good results, erythropoiesis-stimulating agents (ESAs) remain the cornerstone of treatment for anaemia in dialysis patients [8]. However, the individual response to ESAs varies. Some patients may have a poor response, which is defined as ESA resistance and is associated with an increased risk of death [9]. The identification of factors contributing to ESA hyporesponsiveness is crucial to improving MHD patient outcomes. Common factors that cause resistance to ESAs include iron deficiency, chronic inflammation, malnutrition, and secondary hyperparathyroidism. Body mass index (BMI) has been shown to influence the response to ESAs in dialysis patients; however, BMI calculation does not discriminate between fat and lean tissue. Among MHD patients, the impact of body composition on ESA responsiveness is still not well-known [10,11,12]. Bioimpedance methods can be used to determine body composition and may be helpful in controlling anaemia. Based on the abovementioned literature review, the objective of the present study was to explore the effects of body composition on anaemia parameters and ESA responsiveness using the BIS method in MHD patients.

## 2. Materials and Methods

### 2.1. Study Subjects

This cross-sectional study was performed on patients undergoing in-centre MHD at a single facility between January and December 2019. The study was approved by the Institutional Review Board of the Tri-Service General Hospital, and all investigations adhered to the principles of the Declaration of Helsinki. The participants involved in our study gave signed written consent after they were informed about the study procedures, risks, benefits, and their rights. Eligible patients were older than 18 and underwent 4–4.5 h of HD treatments three times each week for at least three months. All patients were dialysed using bicarbonate-based dialysate, high-flux dialysers, and Toray TR-8000 dialysis machines. Blood flow and dialysate flow were 250–350 mL/min and 500–700 mL/min, respectively. Dry weight was clinically determined, considering the intradialytic symptoms and blood pressure, as well as BIS measurement. We excluded patients with active bleeding, acute infectious diseases, or hospitalisation in the three months prior to the start and during the course of the study. Patients with pacemakers, defibrillators, or metallic artificial joints and those who were pregnant or had undergone amputations were excluded to avoid inaccurate BIS measurements.

The following patient data were collected from medical records at the start of the study: age, sex, height, comorbidities, underlying causes of ESRD, dialysis vintage, type of vascular access, and drug medications. Cardiovascular disease was defined as a history of ischaemic heart disease, congestive heart failure, stroke, or peripheral artery disease. Chronic lung disease was defined as a chronic obstructive pulmonary disease requiring long-term treatment with bronchodilators or steroids. Chronic liver disease was defined as chronic viral hepatitis or liver cirrhosis. BMI was calculated as postdialysis weight (kg)/height (m)^2^. Interdialytic weight gain (IDWG) was calculated as (predialysis weight − previous postdialysis weight)/dry weight × 100%. We calculated the mean BMI and IDWG on the day of BIS measurement each month. Residual diuresis was defined as a self-reported urine output of greater than 200 mL daily and was ascertained at baseline and the 12-month follow-up.

### 2.2. Response to ESA

ESA and intravenous iron sucrose were prescribed to maintain a target haemoglobin level within the range of 10 to 11.5 g/L [13]. The route of administration and dose of ESAs treatment were as follows: subcutaneous epoetin beta 2000–6000 IU every week, intravenous darbepoetin 20–40 mcg every week, and intravenous methoxy polyethylene glycol-epoetin β 25–50 mcg every two weeks. The doses were adjusted weekly as necessary according to the patients’ clinical symptoms, body weight, and haemoglobin levels, taking into account of adverse effects of high ESA doses. For those patients receiving darbepoetin and methoxy polyethylene glycol-epoetin β, doses in micrograms were converted to international units by using conversion factors of 200 and 225, respectively [14]. The ESA dose was recorded for each patient as international units administered and was calculated as a weekly dose divided by body weight (IU/kg/week). The erythropoietin resistance index (ERI) was determined as the weekly ESA dose per kg of body weight (IU/kg/week) divided by the haemoglobin level (g/dL) [15]. For each patient, the ESA dose and ERI were calculated each month, and finally, the mean values at 12 months were calculated. The iron dose was recorded as the total dose given throughout the entire course of the study.

### 2.3. Laboratory Measurements

Blood samples were collected monthly before the first dialysis session of the week (Monday or Tuesday). All laboratory data were calculated as the mean of the measurements each month. The biochemical analyses were performed using an automated biochemical analyser (DxC 700 AU Chemistry Analyser, Beckman Coulter, California, USA). The ferritin and intact parathyroid hormone levels were determined by chemiluminescence (ADVIA CentaurCP, Siemens, Munich, Germany). Transferrin saturation (TS) was calculated as serum iron level (μg/dL)/total iron-binding capacity (μg/dL). The Kt/V and normalised protein catabolic rate (nPCR) were calculated using a single-pool urea kinetic model [16]. The parameter Kt/V is a measurement of adequacy of a HD session, where K is the dialyser clearance of urea (L/h), t is the duration of dialysis (h), and V is the distribution volume of urea (L). A previous study has found that a haemoglobin level <10 g/dL was significantly associated with elevated cardiovascular and all-cause mortality [17]. Because the KDIGO guideline suggested a target haemoglobin level of 10 to 11.5 g/L, patients were divided into two groups using a cutoff value of 10 g/dL [13].

### 2.4. Malnutrition–Inflammation Score (MIS)

The MIS is a practical and inexpensive scoring system to assess malnutrition and inflammation in CKD patients [18]. The MIS incorporates four sessions, including the patient’s medical history, physical examination, BMI, and laboratory parameters, and 10 components. The five medical history-based components include dry weight changes, dietary intake, gastrointestinal symptoms, functional capacity, and comorbidity, which includes dialysis vintage. The physical examination comprises decreased fat storage or loss of subcutaneous fat and signs of muscle wasting. Laboratory parameters are serum albumin and total iron-binding capacity. Each component of the score has four levels of severity, ranging from 0 (normal) to 3 (severely abnormal). The sum of all 10 components ranges from 0 (normal) to 30 (severe malnutrition and inflammation). All MIS assessments were performed by the same physician.

### 2.5. Measurement of Body Composition

The BIS assessed using the Body Composition Monitor (BCM, Fresenius Medical Care, Bad Homburg, Germany) was part of the standard of care and was performed every month at our dialysis unit. All measurements were taken by one well-trained nurse after the patients had been in the supine position for at least five minutes before HD treatment. Electrodes were attached to the patient’s contralateral forearm with the arteriovenous fistula or graft and ipsilateral ankle. The BCM measures the body resistance and reactance after applying low-strength alternating electric currents at 50 different frequencies, ranging between 5 and 1000 kHz. Based on the measured resistance and reactance data, ECV, intracellular volume, and total body water were determined using the approach described by Moissl et al. [19].

Overhydration (OH), lean tissue mass, and fat tissue mass were calculated automatically by BCM software (Fresenius Medical Care, Bad Homburg, Germany) according to a three-compartment model [18]. The OH value represented the difference between the normal expected ECV under normal physiological conditions and the measured ECV, whereas relative OH represented the OH value to the ECV ratio. The fat tissue index (FTI) and lean tissue index (LTI) were determined by fat and lean tissue mass adjusted for body surface (kg/m^2^). All BCM measurements, including relative OH, LTI, and FTI, were averaged monthly throughout the follow-up period. The model used in BCM has been extensively validated against gold standards and showed good agreement in HD patients [19,20]. A relative OH value of ≥15% was defined as fluid overload [21].

### 2.6. Statistical Analysis

The Kolmogorov–Smirnov test was performed to check the normality of the data distribution. Continuous variables are expressed as the means ± standard deviations (SDs), and nonnormal variables are expressed as medians and 25th–75th percentiles. Continuous data were compared by two-tailed unpaired Student’s *t* test or the Mann–Whitney *U* test, as appropriate. Categorical data were compared by the χ^2^ test or Fisher’s exact test. Correlations between continuous variables were assessed by Pearson or Spearman coefficients. Multivariate logistic regression analysis was utilised to identify patient factors predicting haemoglobin level <10 g/dL. Stepwise linear regression analysis was used to determine independent factors affecting ESA doses and ERI. A *p* value < 0.05 was considered statistically significant. The data were processed using SPSS version 20 (SPSS Inc., IBM company, New York, NY, USA).

## 3. Results

### 3.1. Patient Characteristics

A total of 77 MHD patients were studied (Figure 1). The sample size provided 100% of power (α = 0.05, two-tail) on haemoglobin, ESA dose, and ERI to detect statistically significant differences between the groups. The baseline characteristics are shown in Table 1. The causes of ESRD were glomerulonephritis in 33.8% of the patients, diabetes mellitus in 58.4%, hypertension in 1.3%, chronic interstitial nephritis in 2.6%, and hereditary polycystic kidney disease in 2.6%. Fifty-one (66.2%) patients were receiving angiotensin-converting enzyme inhibitor/angiotensin II receptor blocker. The mean haemoglobin level was 10.3 ± 1.4 g/dL, and 27 (35.1%) patients had a haemoglobin level <10 g/dL. All patients were treated with ESAs, and 53 (68.8%) patients received epoetin β, 10 (13%) patients received darbepoetin, and 14 (18.2%) received methoxy polyethylene glycol-epoetin β. The mean ERI was 7.0 ± 3.0 IU/week/kg/g/dL. The mean transferring saturation was 24.9 ± 6.9%. Fifty-two (67.5%) patients received treatment with intravenous iron sucrose. Regarding the fluid status and body composition, the mean predialysis-relative OH was 11.1 ± 6.8%. Twenty-one (27.3%) patients had predialysis fluid overload (relative OH value ≥15%).

### 3.2. Associations between Haemoglobin Level and Relevant Parameters

Compared with patients with haemoglobin levels ≥10 g/dL, those with haemoglobin levels <10 g/dL had significantly higher serum ferritin levels (*p* < 0.001), MIS (*p* < 0.001), relative OH (*p* = 0.006), ESA doses (*p* < 0.001), and ERIs (*p* < 0.001) (Table 2). There were no significant differences in BMI, IDWG, LTI, or FTI between the two groups. Regarding iron supplementation, the percentage of patients receiving iron supplementation and the dose used did not differ between the two groups. The univariate logistic regression model revealed that ferritin level, MIS, and relative OH were independently associated with the risk of having a haemoglobin level <10 g/dL. In the multivariate logistic regression model, ferritin level and MIS were the only significant predictors of haemoglobin level <10 g/dL (Table 3). When all continuous variables were divided into two groups using the medium, ferritin levels and MIS were independently and significantly, *p =* 0.011 and 0.015, respectively, associated with haemoglobin levels <10 g/dL (Appendix A).

### 3.3. Associations between ESA Responsiveness and Relevant Parameters

As shown in Table 4, there were significant inverse correlations of ESA dose and ERI with albumin levels (*r* = −0.352 and −0.317, respectively, *p* = 0.002 and 0.005, respectively), BMI (*r* = −0.516 and −0.415, respectively, *p* < 0.001) and LTI (*r* = −0.500 and −0.473, respectively, *p* < 0.001). On the other hand, direct correlations of ESA dose and ERI with age (*r* = 0.339 and 0.320, respectively, *p* = 0.003 and 0.005, respectively), Kt/V (*r* = 0.554 and 0.417, respectively, *p* < 0.001), ferritin levels (*r* = 0.363 and 0.550, respectively, *p* = 0.001 and <0.001, respectively), and MIS (*r* = 0.462 and 0.496, respectively, *p* < 0.001) were identified. When all potentially related variables were included in a multivariate linear regression analysis, ferritin levels (β = 0.260, *p* = 0.003), Kt/V (β = 0.281, *p* = 0.006), BMI (β = −0.351, *p* < 0.001), and LTI (β = −0.230, *p* = 0.019) can be independent predicting determinants for ESA dose (Table 5). Regarding ESA responsiveness, potential influencing factors of ERI were ferritin levels (β = 0.463, *p* < 0.001), BMI (β = −0.275, *p* = 0.002) and LTI (β = −0.214, *p* = 0.029).

## 4. Discussion

This study examines the relationship between body composition, the severity of anaemia, and response to ESAs in an ESRD population undergoing MHD at a single centre. We showed that higher ferritin levels and MIS independently predicted lower haemoglobin levels. The association between body composition and haemoglobin levels was not significant. In addition, lower BMI and LTI and higher ferritin levels were significantly associated with poor response to ESAs.

Excessive fluid accumulation is common in MHD patients because of difficulty in ECV assessment, intradialytic side effects, and increased IDWG. Several studies have shown that MHD patients have fluid overload, with a prevalence ranging from 25% to 46% [4,21]. In our study population, the prevalence of fluid overload was 27.3%, which is in agreement with previously published data. Expanded ECV can affect intravascular and interstitial constituents that cause a rise in plasma volume relative to circulating total red cell mass, leading to haemodilution [22]. In addition to CKD, anaemia due to haemodilution is also common in patients with cirrhosis of the liver and chronic heart failure [2,23]. However, the severity of anaemia was significantly related to markers of inflammation but not fluid overload in our study. MHD patients frequently have multiple sources of inflammation, including dialysis catheters, oxidative stress, gut dysbiosis, retention of uraemic toxins, and dialyser incompatibility [24]. Chronic inflammation is associated with accelerated atherogenesis, protein energy wasting, and anaemia in MHD patients [25]. This result suggests that the relationship among malnutrition, cardiovascular disease, anaemia, fluid overload, and inflammation is complex. Haemodilution may impact the severity of anaemia, but it is not a major factor in MHD patients.

Although haemoglobin levels failed to correlate with BMI in our study population, an inverse correlation was observed between BMI and resistance to ESAs. In line with previous studies, BMI has been shown to correlate negatively with the weekly ESA dose and ERI, suggesting that obesity has a protective effect against anaemia in dialysis patients [26,27,28]. In fact, CKD patients with higher BMIs have a better prognosis [29]. The paradoxical association is attributed to better nutritional status in patients with obesity. Although BMI is strongly correlated with body fat mass, it does not distinguish fat from lean tissue. Note that MHD patients frequently have fluid overload, even after dialysis treatment, and BMI cannot distinguish excessive fluid status. Accordingly, measurements of body composition are necessary to identify which component (body fat mass, lean mass, or hydration) is associated with the response to ESA in MHD patients.

Kotanko et al. developed regression models for the prediction of body composition and found that MHD patients with high absolute total and subcutaneous adipose tissue required lower weekly doses of ESA and had lower ESA resistance [10]. Another study conducted by Vega et al. using BIS to measure body composition in MHD patients showed that higher fat tissue was associated with a better response to ESA; however, there was no association between ERI and LTI [11]. These findings were explained by patients with higher BMI having a lower uraemic load, which has an inhibitory effect on erythropoiesis [10]. Observational studies have shown that the malnutrition–inflammation complex is associated with ESA resistance, and HD patients with higher BMI have a better nutrition status [30,31]. In addition, abdominal fat tissue can produce adipokines, such as leptin, that are associated with the stimulation of erythropoiesis [32].

In contrast, Takata et al. reported different relationships between body composition and ESA responsiveness in MHD patients. They measured skeletal muscle mass by bioimpedance analysis and found that lower muscle mass was associated with a poorer response to ESA [12]. In agreement with this study, we found that LTI, but not OH or FTI, was associated with the response to ESA in MHD patients. Previous research has identified erythropoietin receptors on murine, rat, human myoblasts, and human skeletal muscle tissue [33]. Erythropoietin can stimulate the proliferative response in murine myoblasts [34]. In human skeletal muscle, exercise and hypoxia can induce the release of erythropoietin [35]. Collectively, these findings suggest that muscle mass is associated with the response to ESA. Diagnostic tools using BIS to measure body composition can help to identify MHD patients with low LTI. Treatment targeting sarcopenia may improve ESA responsiveness.

MHD patients are predisposed to iron deficiency because of gastrointestinal bleeding, blood drawing, and blood loss during HD treatment. Since adequate iron stores are essential to achieve target haemoglobin levels, surveillance tests, including assessments of transferrin saturation and serum ferritin, are used to estimate iron stores. However, serum ferritin levels can also be influenced by various underlying conditions, including inflammation, infection, insulin resistance, metabolic syndrome, chronic liver disease, and malnutrition. Hyperferritinemia is a nonspecific finding in routine medical practice, and only 10% of cases are due to iron overload [36]. Our study showed that higher ferritin levels were correlated with higher required doses of ESA and ERI, therefore, suggesting complex underlying conditions in those with ESA hyporesponsiveness.

Because of the good efficacy of ESA and iron supplementation in correcting anaemia in MHD patients, the role of dialysis adequacy on ESA responsiveness may be masked. Inadequate dialysis has been shown to have an impact on the response to ESA in HD patients [37]. The adequacy of the HD dose is measured by the Kt/V or urea reduction ratio. Previous studies have shown an inverse correlation between Kt/V and ESA requirements [38,39]. Our study showed that a higher Kt/V was associated with a higher required dose of ESA; however, the association between Kt/V and ERI was not statistically significant. As small patients and females have a low V, a higher Kt/V should be assessed to achieve adequate dialysis, and the weight-adjusted ESA dose may also be high in these patients. Of note, most of our patients (97%) had a Kt/V value above 1.3, and a further increase in Kt/V is unlikely to improve ESA responsiveness in adequately dialysed patients [40]. In fact, another large multicentre study showed that there was no association between Kt/V and ERI [41].

Our study has a few limitations. First, the number of participants was too small to have strong statistical power. Second, body composition differs between males and females; however, we cannot split the data into subgroups according to sex because of the small study population. Third, patients with metallic implants or amputation were excluded from our study group because the BIS device passes electrical currents through the body, which might have resulted in selection bias. Accordingly, our study population was relatively healthy and may not be reflective of ESRD patients globally in terms of comorbidities, the severity of anaemia, and nutrition status, and hence, it may not be reproducible. However, the prospectively collected data provide good accuracy, and most of our results are consistent with those of previous studies.

## 5. Conclusions

In this study, investigating the relationship between body composition and ESA responsiveness in MHD patients, high MIS and low muscle mass were found to be associated with anaemia severity and ESA resistance, respectively. Considering the use of BIS measurements to identify MHD patients with low muscle mass, nutritional and exercise interventions targeting sarcopenia are warranted in patients with poor responses to ESAs.

## Figures and Tables

**Figure 1 ijerph-19-05704-f001:**
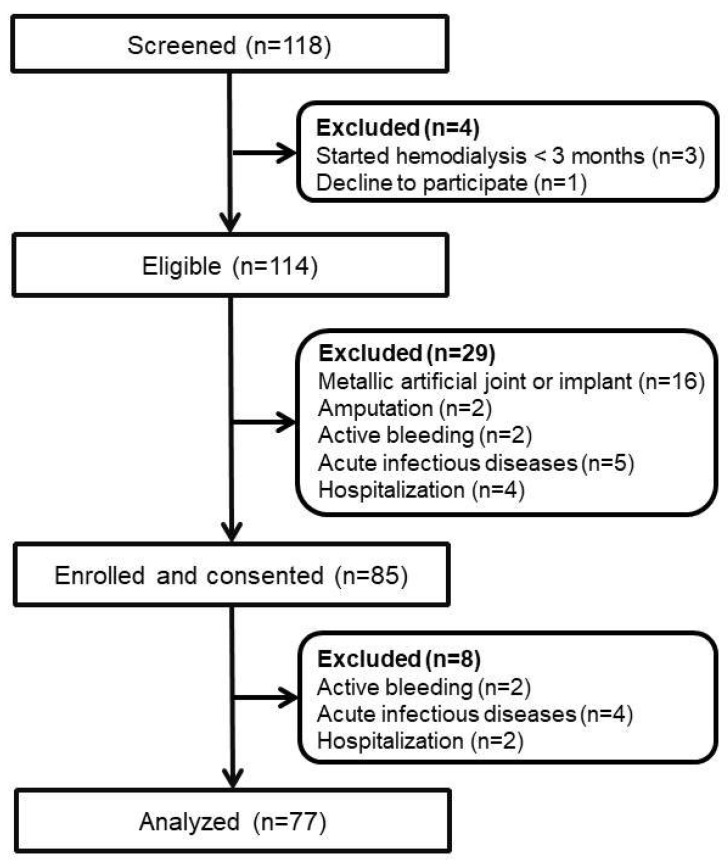
Participants flow chart in this study.

**Table 1 ijerph-19-05704-t001:** Characteristics of the overall study population.

Variable	Data
Sex (% male)	51.9
Age (years)	60.7 ± 12.8
Diabetes mellitus (%)	59.7
Cardiovascular disease (%)	72.7
Chronic lung disease (%)	5.2
Liver disease (%)	11.7
Residual diuresis (%)	23.4
ACEI/ARB (%)	66.2
Dialysis vintage (months)	59.9 (36.3–93.7) *
Vascular access (% AVF)	55.8
Kt/V	1.6 ± 0.2
nPCR (g/kg/day)	1.1 ± 0.2
Haemoglobin (g/dL)	10.3 ± 1.4
MCV (fL)	88.0 ± 7.9
Transferrin saturation (%)	24.9 ± 6.9
Ferritin (ng/mL)	349.2 ± 263.1
Albumin (g/dL)	3.8 ± 0.3
Intact-PTH (pg/mL)	315.9 (151.6–497.9) *
Total cholesterol (mg/dL)	152.6 ± 31.6
MIS	5.3 ± 2.2
BMI (kg/m^2^)	25.8 ± 3.8
IDWG (%)	4.9 ± 2.5
Relative OH (%)	11.1 ± 6.8
LTI (kg/m^2^)	12.9 ± 3.3
FTI (kg/m^2^)	12.0 ± 4.6
Iron dose (mg) ^#^	650.0 ± 372.9
ESA dose (IU/week/kg)	67.8 ± 25.0
ERI (IU/week/kg/g/dL)	7.0 ± 3.0

ACEI/ARB—angiotensin-converting enzyme inhibitor/angiotensin II receptor blocker; AVF—arteriovenous fistula; BMI—body mass index; ERI—erythropoietin resistance index; ESA—erythropoiesis-stimulating agent; FTI—fat tissue index; IDWG—interdialytic weight gain; LTI—lean tissue index; MCV—mean corpuscular volume; MIS—malnutrition–inflammation score; nPCR—normalised protein catabolic rate; OH—overhydration; PTH—parathyroid hormone; * Expressed as the median (interquartile range); ^#^ Patients receiving iron supplementation.

**Table 2 ijerph-19-05704-t002:** Comparison of study patients.

Variable	Haemoglobin < 10 g/dL (N = 27)	Haemoglobin ≥ 10 g/dL (N = 50)	*p*
Sex (% male)	40.7	58.0	NS
Age (years)	64.5 ± 11.6	58.6 ± 13.1	NS
Diabetes mellitus (%)	66.7	56.0	NS
Cardiovascular disease (%)	81.5	68.0	NS
Chronic lung disease (%)	3.7	6.0	NS
Liver disease (%)	18.5	8.0	NS
Residual diuresis (%)	22.2	24.0	NS
ACEI/ARB (%)	63.0	68.0	NS
Dialysis vintage (months)	71.5 (36.1–106.1) *	57.1 (36.0–92.7) *	NS
Vascular access (% AVF)	40.7	64.0	NS
Kt/V	1.6 ± 0.2	1.6 ± 0.3	NS
nPCR (g/kg/day)	1.1 ± 0.2	1.1 ± 0.2	NS
Haemoglobin (g/dL)	8.7 ± 0.8	11.1 ± 0.8	<0.001
MCV (fL)	87.9 ± 10.8	88.0 ± 6.0	NS
Transferrin saturation (%)	24.3 ± 6.1	25.3 ± 7.3	NS
Ferritin (ng/mL)	518.1 ± 341.8	258.0 ± 145.5	<0.001
Albumin (g/dL)	3.7 ± 0.3	3.9 ± 0.3	NS
Intact-PTH (pg/mL)	275.4 (145.5–387.7) *	321.6 (167.8–533.6) *	NS
Total cholesterol (mg/dL)	145.1 ± 38.9	156.6 ± 26.5	NS
MIS	6.5 ± 2.1	4.6 ± 2.1	<0.001
BMI (kg/m^2^)	25.7 ± 3.5	25.8 ± 4.0	NS
IDWG (%)	5.0 ± 2.3	4.8 ± 2.7	NS
Relative OH (%)	13.9 ± 6.4	9.5 ± 6.6	0.006
LTI (kg/m^2^)	11.9 ± 2.9	13.4 ± 3.4	NS
FTI (kg/m^2^)	12.7 ± 4.1	11.7 ± 4.8	NS
Iron supplementation (%)	66.0	70.4	NS
Iron dose (mg) ^#^	778.9 ± 291.7	575.8 ± 398.7	NS
ESA dose (IU/week/kg)	81.5 ± 14.4	60.5 ± 26.5	<0.001
ERI (IU/week/kg/g/dL)	9.5 ± 2.1	5.6 ± 2.5	<0.001

ACEI/ARB—angiotensin-converting enzyme inhibitor/angiotensin II receptor blocker; AVF—arteriovenous fistula; BMI—body mass index; ERI—erythropoietin resistance index; ESA—erythropoiesis-stimulating agent; FTI—fat tissue index; IDWG—interdialytic weight gain; LTI—lean tissue index; MCV—mean corpuscular volume; MIS—malnutrition–inflammation score; nPCR—normalised protein catabolic rate; OH—overhydration; PTH—parathyroid hormone; * Expressed as the median (interquartile range); ^#^ Patients receiving iron supplementation.

**Table 3 ijerph-19-05704-t003:** Predictors of haemoglobin levels <10 g/dL.

Variables	Univariate			Multivariate		
	b	OR (95% CI)	*p*	b	OR (95% CI)	*p*
Age	0.04	1.04 (0.99−1.09)	0.060			
Sex	−0.70	0.50 (0.19−1.29)	0.151			
Diabetes mellitus	−0.45	0.64 (0.24−1.69)	0.364			
Ferritin	0.01	1.01 (1.00−1.01)	0.001	0.01	1.01 (1.00−1.01)	0.004
MIS	0.41	1.50 (1.17−1.93)	0.001	0.32	1.38 (1.04−1.83)	0.025
BMI	−0.01	0.99 (0.88−1.12)	0.906			
Relative OH	0.10	1.11 (1.03−1.20)	0.010	0.06	1.06 (1.00−1.16)	0.206
LTI	−0.15	0.86 (0.74−1.01)	0.059			
FTI	0.05	1.05 (0.95−1.17)	0.329			

BMI—body mass index; FTI—fat tissue index; LTI—lean tissue index; MIS—malnutrition–inflammation score; OH—overhydration; b—logistic regression coefficients; OR—Odds Ratio.

**Table 4 ijerph-19-05704-t004:** Bivariate correlations of ESA responsiveness with relevant parameters.

Variable	ESA Dose		ERI	
	*r*	*p*	*r*	*p*
Age	0.339	0.003	0.320	0.005
Dialysis vintage	−0.090	NS	−0.099	NS
Kt/V	0.554	<0.001	0.417	<0.001
nPCR	0.149	NS	0.102	NS
MCV	0.069	NS	0.007	NS
Transferrin saturation	−0.103	NS	−0.098	NS
Ferritin	0.363	0.001	0.550	<0.001
Albumin	−0.352	0.002	−0.317	0.005
Intact-PTH	−0.182	NS	−0.170	NS
Total cholesterol	0.101	NS	−0.035	NS
MIS	0.462	<0.001	0.496	<0.001
BMI	−0.516	<0.001	−0.415	<0.001
IDWG	0.091	NS	0.078	NS
Relative OH	0.112	NS	0.180	NS
LTI	−0.500	<0.001	−0.473	<0.001
FTI	−0.076	NS	−0.019	NS
Iron dose	0.161	NS	0.162	NS

BMI—body mass index; ERI—erythropoietin resistance index; ESA—erythropoiesis-stimulating agent; FTI—fat tissue index; IDWG—interdialytic weight gain; LTI—lean tissue index; MCV—mean corpuscular volume; MIS—malnutrition–inflammation score; nPCR—normalised protein catabolic rate; OH—overhydration; PTH—parathyroid hormone.

**Table 5 ijerph-19-05704-t005:** Multivariate linear regression analysis predicting determinants of ESA responsiveness.

Variable	ESA Dose		ERI	
	β	*p*	β	*p*
Ferritin	0.260	0.003	0.463	<0.001
Kt/V	0.281	0.006	0.181	0.071
BMI	−0.351	<0.001	−0.275	0.002
LTI	−0.230	0.019	−0.214	0.029

BMI—body mass index; ERI—erythropoietin resistance index; ESA—erythropoiesis-stimulating agent; LTI—lean tissue index.

## Data Availability

The data presented in this study are available on request from the corresponding author.

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
