# Peer review of "Investigation of the Relationship between Lean Muscle Mass and Erythropoietin Resistance in Maintenance Haemodialysis Patients: A Cross-Sectional Study"

_ijerph, 2022, doi:10.3390/ijerph19095704_

Round 1
Reviewer 1 Report
Thank you for exploring such an important topic in medicine that at times becomes so difficult to manage. I have a few comments and suggestions that I find would be useful to answer:
Pages 58-67: Here you are talking about methods of estimating fluid status in HD patients. You mentioned bromide dilution method, and rightfully so, as well as bio-impedance devices, and use of ultrasound. IVC measurement may be more useful when evaluating volume overload rather than euvolemia or hypovolemia, and additionally can be negatively affected by factors, such as pulmonary HTN, that are common in ESRD patients and presence of AVFs. You do not seem to mention the usefulness of right heart catheterization and obtaining parameters, such as RA pressure, PASP, and PCWP, among others to determine volume status in such patients. Would be worth exploring this further and commenting as to what are the drawbacks of this method to be used routinely (e.g., availability of trained personnel, invasive nature with possible complications, costs, etc.)
Line 130 - I understand that nephrologists will be the primary audience of this study, but would be useful to define Kt/V for other healthcare practitioners who are not aware of these details unique to HD patients. Although, I saw this was defined later in discussion section.
2.4 - MIS - does not need to be bolded
3.1 Patient characteristics
I find it very unusual that only <2% of patients had nephropathy attributed to hypertension, which is the top 2 cause of renal disease along with DM. Any particular explanation of this? This is not likely to be related or affecting the results of your study, but just an interesting finding
If all patients were treating with ESA, you only mention what 24 patients were getting specifically. What did the other 53 use?
Based on the data provided, it is worth noting that you probably had a rather "healthy" population of ESRD patients. This may indeed affect the validity and generalizability of your study findings. It is unusual to see that actually most of the parameters were within the target range at baseline for most of the patients.
3.2
After multivariate regression analysis you mentioned ferritin was significantly associated with low Hgb level. Even though statistically this is correct, OR 95% CI crosses 1, which makes it not a risk factor associated with the outcome. Based on analysis, only MIS remains with elevated OR with 95% CI >1 and p value that is <0.05. More so, it is very likely that ferritin level (as an inflammatory marker) and MIS are reflective of the same pathophysiology.
3.3
Why are we even looking at serum iron is isolation? It is not even a marker of iron overload/deficiency and a rather "useless" test in isolation if not used for saturation calculations.
Did MIS lose significance with regard to ESA dose/ERI in multivariate regression? This is interesting as would have expected it to be still relevant, leading to increase ESA resistance. Even though ferritin continued to have positive correlation, and as above MIS and ferritin levels likely to be reflective of the same process.
4. Discussion
Line 249 - as above, disagree that ferritin level was significantly associated with low Hgb level.
In limitations, I would include that your study population may not be reflective of ESRD patients globally in terms of comorbidities, degree of anemia, etc. and hence, may not be reproducible.
Otherwise, a very well written article. Congratulations.
Author Response
Response to Reviewer 1
We greatly appreciated your positive comments on our manuscript. The itemized responses to your comments are below. The changes have highlighted yellow.
- Here you are talking about methods of estimating fluid status in HD patients. You mentioned bromide dilution method, and rightfully so, as well as bio-impedance devices, and use of ultrasound. IVC measurement may be more useful when evaluating volume overload rather than euvolemia or hypovolemia, and additionally can be negatively affected by factors, such as pulmonary HTN, that are common in ESRD patients and presence of AVFs. You do not seem to mention the usefulness of right heart catheterization and obtaining parameters, such as RA pressure, PASP, and PCWP, among others to determine volume status in such patients. Would be worth exploring this further and commenting as to what are the drawbacks of this method to be used routinely (e.g., availability of trained personnel, invasive nature with possible complications, costs, etc.)
Remedy: We agree with this important comment. The following sentences have been added:
“Right heart catheterization provides measurements of hemodynamic data, including right atrium pressure, pulmonary arterial pressure, and pulmonary capillary wedge pressure, and thus can be an indicator of volume status. However, it is an expensive and invasive procedure requiring trained personnel and may be uncommonly complicated with carotid artery injury, arteriovenous fistula formation, and tricuspid valve injury. (please see lines 62–67)
The following reference has been added:
“6. Chen, Y.; Shlofmitz, E.; Khalid, N.; Bernardo, N.L.; Ben-Dor, I.; Weintraub, W.S.; Waksman, R. Right Heart Catheterization-Related Complications: A Review of the Literature and Best Practices. Cardiol Rev 2020, 28, 36-41, doi:10.1097/CRD.0000000000000270.” (please see lines 393–395)
- I understand that nephrologists will be the primary audience of this study, but would be useful to define Kt/V for other healthcare practitioners who are not aware of these details unique to HD patients. Although, I saw this was defined later in discussion section.
Remedy: Thank you for reminding us of this important point. The following sentences have been added:
“The parameter Kt/V is a measurement of adequacy of a hemodialysis session, where K is the dialyzer clearance of urea (L/hr), t is duration of dialysis (hr), and V is the distribution volume of urea (L).” (please see lines 142–144)
- MIS - does not need to be bolded
Remedy: Thank you for this comment. The following sentence has been revised:
“Malnutrition-inflammation score (MIS)” (please see line 148)
- I find it very unusual that only <2% of patients had nephropathy attributed to hypertension, which is the top 2 cause of renal disease along with DM. Any particular explanation of this? This is not likely to be related or affecting the results of your study, but just an interesting finding
Remedy: Thank you for this important comment. In our study population, the cause of ESRD was glomerulonephritis in 26 patients. Most of them (22 patients) received a diagnosis based on kidney biopsy, including chronic glomerulonephritis, minimal change disease, focal segmental glomerulosclerosis, and IgA nephropathy. All of these 22 patients also had hypertension. If they did not undergo kidney biopsy, they might be categorized as having hypertensive nephropathy clinically.
- If all patients were treating with ESA, you only mention what 24 patients were getting specifically. What did the other 53 use?
Remedy: Thank you for reminding us of this important point. The following sentence has been revised:
“All patients were treated with ESAs; 53 (68.8%) patients received epoetin β, 10 (13%) patients received darbepoetin, and 14 (18.2%) received methoxy polyethylene glycol-epoetin β.” (please see line 205)
- Based on the data provided, it is worth noting that you probably had a rather "healthy" population of ESRD patients. This may indeed affect the validity and generalizability of your study findings. It is unusual to see that actually most of the parameters were within the target range at baseline for most of the patients.
Remedy: Thank you very much for this invaluable comment. In our study, patients with pacemakers, defibrillators, or metallic artificial joints and those who had undergone amputations were excluded to avoid inaccurate BIS measurements. These patients had higher frailty and were more likely to have poor health and abnormal laboratory test results. Therefore, the exclusion criteria may affect the validity and generalizability of our study findings.
- After multivariate regression analysis you mentioned ferritin was significantly associated with low Hgb level. Even though statistically this is correct, OR 95% CI crosses 1, which makes it not a risk factor associated with the outcome. Based on analysis, only MIS remains with elevated OR with 95% CI >1 and p value that is <0.05. More so, it is very likely that ferritin level (as an inflammatory marker) and MIS are reflective of the same pathophysiology.
Remedy: We appreciate the invaluable comment. After multivariate regression analysis, the ferritin OR 95% CI was 1.002–1.009 that did not cross 1. When all continuous variables were divided into two groups using medium, ferritin level and MIS remained to be the significant predictors of hemoglobin level < 10 g/dL. (please see supplement table 1) These results demonstrated that higher ferritin levels and MIS independently predicted lower hemoglobin levels, suggesting the effect of chronic inflammation on anemia in MHD patients. The following sentence has been added:
“When all continuous variables were divided into two groups using medium, ferritin level and MIS remained to be the significant predictors of haemoglobin level < 10 g/dL (Supplement table 1).” (please see lines 230–232)
- Why are we even looking at serum iron is isolation? It is not even a marker of iron overload/deficiency and a rather "useless" test in isolation if not used for saturation calculations
Remedy: Thank you for reminding us of this important point. The following sentence has been revised:
“As shown in Table 4, there were significant inverse correlations of ESA dose and ERI with serum iron (r = −0.347 and −0.333, respectively, p < 0.05), albumin levels (r = −0.352 and −0.317, respectively, p < 0.05), BMI (r = −0.516 and −0.415, respectively, p < 0.001) and LTI (r = −0.500 and −0.473, respectively, p < 0.001).” (please see line 246)
- Did MIS lose significance with regard to ESA dose/ERI in multivariate regression? This is interesting as would have expected it to be still relevant, leading to increase ESA resistance. Even though ferritin continued to have positive correlation, and as above MIS and ferritin levels likely to be reflective of the same process.
Remedy: We appreciate the invaluable comment. MIS was a significant predictor of ESA dose and ERI in simple linear regression however it was not significant in multivariate regression analysis. As your comment, our study population was relatively healthy, and the effect of chronic inflammation on ESA responsiveness might be blunted.
- As above, disagree that ferritin level was significantly associated with low Hgb level.
Remedy: Thank you for this comment. As mention above, higher ferritin levels and MIS independently predicted lower haemoglobin levels, suggesting the effect of chronic inflammation on anemia in MHD patients.
- In limitations, I would include that your study population may not be reflective of ESRD patients globally in terms of comorbidities, degree of anemia, etc. and hence, may not be reproducible.
Remedy: Thank you for this important comment. The following sentences have been added:
“Accordingly, our study population was relatively healthy and may not be reflective of ESRD patients globally in terms of comorbidities, severity of anemia, and nutrition status, and hence, may not be reproducible.” (please see lines 352–355)
Lastly, we are deeply honored by the time and effort you spent in reviewing this manuscript. By incessantly reviewing and revising our test, we were spurred to read more and thus learn more from your criticisms.

Reviewer 2 Report
In this manuscript, the author has studied the relationship between body composition and ESA responsiveness in MHD patients. Eventually, the author has concluded that high MIS and low muscle mass are associated with anemia severity and ESA resistance, respectively. I have some questions below.
- The sample volume was 77 patients. Is this sample volume enough? Generally, people use a sample volume of over one hundred.
- Does the ESA treatment the same for all patients? If not, how does the author address the variant between patients?
- In table 2, the author has set a hemoglobin line that is ten g/dL. Please explain the rationality.
- What is the unit of table 3? What do OR and B mean?
- Still the table problem. In Table 4, the serum iron for ESA dose and ERI were r = −0.347 and −0.333, respectively. I thought the P-values were calculated from each group. However, in section 3.3, the author mentioned p<0.05. How did the author calculate it?
- The conclusion 'When all potentially related variables were included in a multivariate linear regression analysis, predictors of higher ESA does were ferritin levels, BMI, and LTI.' is not clear.
Author Response
Response to Reviewer 2
We greatly appreciated your important comments on our manuscript. The itemized responses to your comments are below.
- The sample volume was 77 patients. Is this sample volume enough? Generally, people use a sample volume of over one hundred.
Remedy: Thank you for this important comment. The following sentence has been added:
“The sample size provided 100% of power (α = 0.05, two-tail) on haemoglobin, ESA dose and ERI to detect statistically significant differences between the groups.” (please see lines 197–199)
- Does the ESA treatment the same for all patients? If not, how does the author address the variant between patients?
Remedy: We appreciate the invaluable comment. The ESA treatment was the same for all patients. ESA and intravenous iron sucrose were prescribed to maintain a target haemoglobin level within the range of 10 to 11.5 g/L. The following sentences have been added:
“The route of administration and dose of ESAs treatment were as follows: subcutaneous epoetin beta 2000–6000 IU every week, intravenous darbepoetin 20–40 mcg every week, and intravenous methoxy polyethylene glycol-epoetin β 25–50 mcg every two weeks. The doses were adjusted weekly as necessary according to patient’s clinical symptoms, body weight, and haemoglobin level, taking into account of adverse effects of high ESA doses.” (please see lines 118–123)
- In table 2, the author has set a hemoglobin line that is ten g/dL. Please explain the rationality.
Remedy: Thank you for this important comment. Because the KDIGO guideline suggested a target haemoglobin level of 10 to 11.5 g/L, patients were divided into two groups using a cut-off value of 10 g/dL. The following sentence has been added:
“Previous study has found a hemoglobin level <10 g/dL was significantly associated with elevated cardiovascular and all‐cause mortality.” (please see lines 144–145)
The following reference has been added:
“17. Kuo, K.L.; Hung, S.C.; Tseng, W.C.; Tsai, M.T.; Liu, J.S.; Lin, M.H.; Hsu, C.C.; Tarng, D.C.; Taiwan Society of Nephrology Renal Registry Data, S. Association of Anemia and Iron Parameters With Mortality Among Patients Undergoing Prevalent Hemodialysis in Taiwan: The AIM - HD Study. J Am Heart Assoc 2018, 7, e009206, doi:10.1161/JAHA.118.009206.” (please see lines 419–421)
- What is the unit of table 3? What do OR and B mean?
Remedy: Thank you for reminding us of this important point. The units of all variables have been added. The following footnote has added:
“b, logistic regression coefficients; OR, Odds Ratio” (please see line 241)
- Still the table problem. In Table 4, the serum iron for ESA dose and ERI were r = −0.347 and −0.333, respectively. I thought the P-values were calculated from each group. However, in section 3.3, the author mentioned p<0.05. How did the author calculate it?
Remedy: We agree with this important comment. We have revised the following sentence: “As shown in Table 4, there were significant inverse correlations of ESA dose and ERI with serum iron (r = −0.347 and −0.333, respectively, p = 0.002 and 0.003, respectively), albumin levels (r = −0.352 and −0.317, respectively, p = 0.002 and 0.005, respectively), BMI (r = −0.516 and −0.415, respectively, p < 0.001) and LTI (r = −0.500 and −0.473, respectively, p < 0.001). On the other hand, direct correlations of ESA dose and ERI with age (r = 0.339 and 0.320, respectively, p = 0.003 and 0.005, respectively), Kt/V (r = 0.554 and 0.417, respectively, p < 0.001), ferritin levels (r = 0.363 and 0.550, respectively, p = 0.001 and < 0.001, respectively), and MIS (r = 0.462 and 0.496, respectively, p < 0.001) were identified.” (please see line 4–6). In response to Reviewer 1, we have deleted “serum iron (r = −0.347 and −0.333, respectively, p = 0.002 and 0.003, respectively)”. (please see lines 246–251)
- The conclusion 'When all potentially related variables were included in a multivariate linear regression analysis, predictors of higher ESA does were ferritin levels, BMI, and LTI.' is not clear.
Remedy: Thank you very much for this invaluable comment. We have revised the following sentence:
“When all potentially related variables were included in a multivariate linear regression analysis, ferritin levels (b = 0.260, p = 0.003), Kt/V (b = 0.281, p = 0.006), BMI (b = −0.351, p < 0.001), and LTI (b = −0.230, p = 0.019) can be independent predicting determinants for ESA dose.” (please see lines 252–255)
Lastly, we are deeply honored by the time and effort you spent in reviewing this manuscript. By incessantly reviewing and revising our test, we were spurred to read more and thus learn more from your criticisms.

Round 2
Reviewer 2 Report
No more questions.